

# Comparative transcriptomic analysis reveals the cold acclimation during chilling stress in sensitive and resistant passion fruit (*Passiflora edulis)* cultivars

Yanyan Wu[1], Weihua Huang[1], Qinglan Tian[1], Jieyun Liu[1], Xiuzhong Xia[2], Xinghai Yang[2] and Haifei Mou[1]

[1] Biotechnology Research Institute, Guangxi Academy of Agricultural Sciences, Nanning 530007, Guangxi, China

[2] Rice Research Institute, Guangxi Academy of Agricultural Sciences, Nanning 530007, Guangxi, China

## ABSTRACT

Chilling stress (CS) is an important limiting factor for the growth and development of passion fruit (*Passiflora edulis*) in winter in South China. However, little is known about how the passion fruit responds and adapts to CS. In this study, we performed transcriptome sequencing of cold-susceptible cultivar Huangjinguo (HJG) and cold-tolerant cultivar Tainong 1 (TN1) under normal temperature (NT) and CS conditions, and a total of 47,353 unigenes were obtained by seven databases. Using differentially expressed unigenes (DEGs) analysis, 3,248 and 4,340 DEGs were identified at two stages, respectively. The Gene Ontology (GO) enrichment analysis showed that the DEGs were mainly related to phosphorylation, membrane protein, and catalytic activity. In Kyoto Encyclopedia of Genes and Genomes (KEGG) pathway, the unigenes of plant-pathogen interaction, plant hormone signal transduction and fatty acid metabolism were enriched. Then, the 12,471 filtered unigenes were clustered into eight co-expression modules, and two modules were correlated with CS. In this two modules, 32 hub unigenes were obtained. Furthermore, the unigenes related to CS were validated using quantitative real-time PCR (RT-qPCR). This work showed that the expression levels of CS-related unigenes were very different in two passion fruit cultivars. The results provide information for the development of passion fruit with increased chilling tolerance.

# INTRODUCTION

Passion fruit is a tropical and subtropical fruit tree that is widely planted in South China and its fruit has an aromatic smell and high nutritional values. But passion fruit is susceptible to cold stress in winter (*Liu et al., 2017a*), which can cause large economic loss.

Cold stress is one of the limiting factors for plant growth and development (*Shi, Ding & Yang, 2018*). In plants, cold stress is classified into CS (0−15 °C) and freezing stress (<0 °C) (*Yadav, 2010*; *Shi, Ding & Yang, 2018*). The cold environment can cause changes in the structure and activity of proteins in plant cells, leading to altered enzymatic reactions

Corresponding authors
Xinghai Yang,
yangxinghai888@gxaas.net,
yangxinghai514@163.com
Haifei Mou, mhf@gxaas.net

such as photosynthesis and respiration, and eventually leading to symptoms such as wilting and yellowing of plant leaves (*Hendrickson et al., 2006*). When plants are in reproductive growth, cold stress can cause damage of the plant reproductive organs, and the seed setting rate will be significantly reduced, which will eventually affect crop yields and cause major losses to agricultural production. Plants can gain resistance to low temperature, and this process is called cold acclimation. The cold acclimation of plants includes changes in a variety of intracellular physiological and biochemical processes. The most significant changes include the instantaneous increase of calcium ion concentration (*Carpaneto et al., 2007*), growth cessation, decrease in tissue water content affects on plant hormones abscisic acid (ABA), brassinolide (BR), and gibberellin (GA), cause fatty acid unsaturation and lipid peroxidation (*Hara et al., 2003*), changes in phospholipid composition (*Webb, Uemura & Steponkus, 1994*), and penetrating substance such as proline, betaine and soluble sugar (*Krasensky & Jonak, 2012*). The molecular mechanism of cold acclimation is that non-freezing low temperature can induce plants to express a series of cold response proteins to help plants resist freezing at low temperature. Inducer of CBF expression (ICE)- C-repeat binding factors (CBF)- cold-regulated proteins (COR) is thought to be one of the most important defense pathways in plants against cold stress (*Shi, Ding & Yang, 2018*). CBF can regulate the expression of COR by binding to the C-repeat/dehydration-responsive element (CRT/DRE) sequence that resides in the promoter region of the *COR* gene (*Stockinger, Gilmour & Thomashow, 1997*; *Liu et al., 1998*). ICE1 is located upstream of *CBF*, and it is a MYC-like bHLH type transcription factor, which can bind to the recognition site of the *CBF3* promoter and regulate its expression (*Chinnusamy et al., 2003*) Moreover, some CBF-independent transcription factors are involved in modulating COR expression, and various transcription factors, including CAMTA3 (*Doherty et al., 2009*), ZAT12 (*Vogel et al., 2005*), and HY15 (*Catalá, Medina & Salinas, 2011*) can regulate the expression of CBFs. Protein phosphorylation also plays an important role in regulating the response of plants to low temperature (*Mann, 2003*), and mitogen-activated protein kinase (MAPK) as an important element in signal transmission (*Zhao et al., 2017*).

Guangxi belongs to a tropical and subtropical monsoon climate. The coldest month in January has an average daily temperature of 5.5 °C to 15.2 °C. The continuous low temperature in winter affects the growth of passion fruit. However, no systematic study on the CS of passion fruit has been reported. In this study, the RNA-seq data was used to analyze gene expression during CS in passion fruit cultivars HJG and TN1. The main aims are to (i) analyze the gene expression profile of passion fruit during CS, (ii) explore the functions of DEGs, (iii) construct regulation network of the interactions of chilling tolerance genes of passion fruit, and (iv) identify the hub genes that affect the CS of passion fruit.

## MATERIALS AND METHODS

### Plant materials

HJG was introduced in the Bannahuangguo of Xishuangbanna Botanical Garden in Yunnan, and it is a cold-sensitive accession. TN1 comes from Taiwan, and it is a cold-resistant purple passion fruit. The cutting seedling heights ranged from 29 to 38 cm, and

the seedlings were transplanted in Nanning experimental field (Guangxi, China, 22.85°N, 108.26°E) on May 25, 2019. The first sampling time was November 25, 2019, at 10 a.m., and the temperature was 25 °C. The second sampling time was January 18, 2020, at 10 a.m., and the temperature was 7 °C. The fresh leaves of passion fruit were snap frozen in liquid nitrogen, and then stored in −80 °C freezer. Each sample had three biological replicates. Under NT condition, the three biological replicates of HJG, HJGA1, HJGA2 and HJGA3, were recorded as A1; the three biological replicates of TN1, TN1A1, TN1A2 and TN1A3 were recorded as A2. Under CS condition, three biological replicates of HJG, HJGB1, HJGB2 and HJGB3, were denoted as B1; three biological replicates of TN1, TN1B1, TN1B2 and TN1B3, were denoted as B2.

### RNA extraction, sequencing, assembly and annotation

Total RNA was extracted with RNAprep Pure kit (Tiangen, Beijing, China) according to the manufacturer's instructions. Nanodrop2000 (Shimadzu, Japan) was used to detect the concentration and purity of the extracted RNA. Agarose gel electrophoresis was used to detect the integrity of the RNA, and Agilent 2100 (Agilent, America) was used to determine the RIN value. A single library requires 1 μg of RNA, with a concentration of ≥ 50 ng/μL, and OD260/280 between 1.8 and 2.2. Magnetic beads (Invitrogen, America) with Oligo (dT) was used to pair with the 3′ poly A tail of eukaryotic mRNA, thus isolating mRNA from total RNA. Subsequent, reverse synthesis of cDNA was performed. These libraries above were sequenced using an Illumina NovaSeq 6000 sequencer (Illumina Inc., USA) and 150-bp paired-end reads were generated. In order to ensure the accuracy of subsequent analysis, the original sequencing data were filtered first to obtain clean data.

We used Trinity v2.8.6 (*Haas et al., 2013*) to splice the transcript fragments to obtain transcripts, and then used CD-HIT to cluster the transcript sequences to remove redundant sequences and get all the unigene sequence sets for the subsequent analysis. Bowtie 2 (*Salzberg & Langmead, 2012*) was used to align the sequencing data to the reconstructed unigene sequence set, and the alignment file was mainly used for subsequent unigene quantification and differential expression analysis. The unigene sequences were compared with the NCBI non-redundant protein sequences (NR), Swiss-Prot, TrEMBL, KEGG, GO, Pfam, and EuKaryotic Orthologous Groups (KOG) databases using Basic Local Alignment Search Tool (BLAST). Finally, HMMER3 (*Finn, Clements & Eddy, 2011*) was used to align the amino acid sequence of unigene with the Pfam database to obtain the annotation information of unigenes.

### Enrichment analysis of DEGs

The read counts and transcripts per million reads (TPM ) were calculated using RSEM (*Li & Dewey, 2011*) and bowtie2 (*Salzberg & Langmead, 2012*). The DEGs were identified through the software packages of Bioconductor 3.11-DESeq2 (*Love, Huber & Anders, 2014*). The screening threshold is false discovery rate (FDR ) <0.05, and log2 fold change (FC (condition 2/condition 1) for a gene) >1 or log2FC <−1. The DEGs were classified, and GO and KEGG enrichment analysis were subsequently performed.

 

## Weighted gene co-expression network analysis

We followed these steps below for weighted gene co-expression network analysis (WGCNA): (i) screening DEGs for WGCNA cluster analysis; (ii) calling the R package to cluster the DEGs; (iii) calling ggplot2 in the R package to draw the clustering heat map and histogram of each module; (iv) using the top GO terms to perform GO enrichment analysis on each module; (v) calling Fisher-test function in R for KEGG enrichment analysis; and (vi) using Cytoscape3.8.0 (*Su et al., 2014*) to draw network diagram. A signed network was constructed using the blockwiseModules function, with the following parameters: power = 14, minModuleSize = 30, mergeCutHeight = 0.25, corType = pearson. When co-expressed genes are defined according to the above-mentioned standards, each gene is assigned a module number and corresponding module color; otherwise the 'gray' module was used.

## Validation of the CS-related genes using RT-qPCR

We selected 11 genes related to plant hormone signaling, fatty acid metabolism and plant-pathogen interaction using GO and KEGG databases, and 4 hub genes in WGCNA for validation. The primers were designed using Primer3 (Table S1). Using *HIS* as the reference gene (*Liu et al., 2017A*), RT-qPCR was used to analyze the expression levels of 15 genes in B1 and B2.

The identical RNA samples as RNA-seq experiments were used for RT-qPCR analysis. The detailed experimental method refers to *Wu et al. (2020)*. The relative gene expression level was calculated by reference to the $2^{-\Delta\Delta Ct}$ method (*Livak & Schmittgen, 2001*). All unigenes expression analysis were performed in triplicates. The values represented arithmetic averages of three replicates, and the data were expressed as a mean plus and minus standard deviation (mean $\pm$ SD).

## Statistical analysis

CASAVA was used for base calling. Subsequently, we used SeqPrep for quality control of raw sequencing data. Pearson correlation coefficient is used to measure the correlation between samples. The package heatmap of R was used to prepare the correlation between samples and DEGs expression pattern clustering. Data of RT-qPCR was analyzed using Excel 2007. The figures were prepared using Origin 9.65.

# RESULTS

## Quality control and assembly of passion fruit transcriptome sequences

To compare gene expression profiles of the two passion fruit cultivars under NT and CS, transcriptome sequencing and analysis were performed. After decontamination and adaptor removal, 533,935,574 raw reads were obtained from 12 samples (NCBI accession number: PRJNA634206), a total of 80.09 Gb clean reads and 6.67 Gb per sample. The Q30 base percentage was 93.22% and GC content was 44.64% (Table 1).

The clean reads were assembled into transcripts using the Trinity in paired-end method, and 211,874 transcripts were obtained (https://doi.org/10.6084/m9.figshare.13489869.v2).

**Table 1  Statistical results of transcriptome sequencing.**

| Sample | Reads number | Total base (bp) | Q30 (%) | GC content (%) |
|---|---|---|---|---|
| HJGA1 | 46599672 | 6989950800 | 93.02 | 45.36 |
| HJGA2 | 44522400 | 6678360000 | 93.08 | 44.49 |
| HJGA3 | 46642622 | 6996393300 | 93.23 | 45.31 |
| TN1A1 | 47166760 | 7075014000 | 93.18 | 45.23 |
| TN1A2 | 43505726 | 6525858900 | 93.31 | 44.33 |
| TN1A3 | 45058566 | 6758784900 | 93.16 | 43.83 |
| HJGB1 | 44758052 | 6713707800 | 93.34 | 43.83 |
| HJGB2 | 43843522 | 6576528300 | 93.43 | 44.70 |
| HJGB3 | 39087142 | 5863071300 | 93.14 | 43.37 |
| TN1B1 | 49306634 | 7395995100 | 93.31 | 45.51 |
| TN1B2 | 44267228 | 6640084200 | 93.17 | 44.87 |
| TN1B3 | 39177250 | 5876587500 | 93.27 | 44.89 |

**Table 2  Sequencing data mapped to unigene set.**

| Sample | Pair reads | Aligned concordantly 0 times | Aligned concordantly exactly 1 time | Aligned concordantly >1 times | Total alignment ratio (%) |
|---|---|---|---|---|---|
| HJGA1 | 23299836 | 6183910 | 15486803 | 1629123 | 83.19 |
| HJGA2 | 22261200 | 6461646 | 14313504 | 1486050 | 81.73 |
| HJGA3 | 23321311 | 7002139 | 14768806 | 1550366 | 81.04 |
| TN1A1 | 23583380 | 8597970 | 13025983 | 1959427 | 75.69 |
| TN1A2 | 21752863 | 7919003 | 11974898 | 1858962 | 75.79 |
| TN1A3 | 22529283 | 7817813 | 12764435 | 1947035 | 77.11 |
| HJGB1 | 22379026 | 6946292 | 14131411 | 1301323 | 78.91 |
| HJGB2 | 21921761 | 6859052 | 13821399 | 1241310 | 78.45 |
| HJGB3 | 19543571 | 5267390 | 12862365 | 1413816 | 81.72 |
| TN1B1 | 24653317 | 9000336 | 13737412 | 1915569 | 74.21 |
| TN1B2 | 22133614 | 8908390 | 11548761 | 1676463 | 70.83 |
| TN1B3 | 19588625 | 6676080 | 11320050 | 1592495 | 75.95 |

The CD-HIT was then used to cluster the transcripts, which yielded 47,353 unigenes (https://doi.org/10.6084/m9.figshare.13489863.v4) with a mean length of 1,211 bp, N50 length of 2,368 bp, and N90 of 450 bp. Afterwards, Bowtie2 was used to align the sequences of each sample to the unigene sequence set, with an average alignment ratio of 77.89% (Table 2).

## Unigene function annotation

The assembled unigenes were annotated to databases including the NR, Swiss-Prot, TrEMBL, KEGG, GO, Pfam, and KOG, to which 97.92%, 70.40%, 97.82%, 33.97%, 36.16%, 61.43%, and 48.92% of unigenes were mapped, respectively. A total of 47,353

**Table 3 Unigenes were annotated to seven databases.**

| Database | Annotated number | Annotated ratio (%) |
| --- | --- | --- |
| GO | 17123 | 36.16 |
| KEGG | 16086 | 33.97 |
| KOG | 23164 | 48.92 |
| NR | 46369 | 97.92 |
| Pfam | 29091 | 61.43 |
| Swiss-Prot | 33337 | 70.40 |
| TrEMBL | 46323 | 97.82 |
| Total | 47353 | 100 |

unigenes acquired annotation information (Table 3). The number of annotated unigenes in NR and TrEMBL was the largest, which were 46,369 and 46,323, respectively.

In the GO database, 17,123 unigenes were annotated and matched to three major categories: biological process (BP), cellular component (CC) and molecular function (MF). Enriched BP terms were mainly about "metabolic process" (4,350) and "cellular process" (2,191). Enriched CC terms were mainly about "membrane part" (1,270) and "cell part" (890). Enriched MF terms were mainly about "binding" (7,367) and "catalytic activity" (5,715) (Fig. 1A).

In KOG database, 23,164 unigenes were annotated, which were clustered into 25 categories. The unigenes were mainly about "signal transduction mechanisms" (2,439) and "posttranslational modification, protein turnover, chaperones" (2,138) (Fig. 1B).

In KEGG database, 16,086 unigenes were annotated. According to the functions, these unigenes were enriched in 9 pathways. The enriched pathways were mainly about "metabolism" (10,045) and "organismal systems" (4,505) (Fig. 1C).

## Comparative analysis of DEGs in two cultivars at two stages

In order to gain insights on the adaptation of passion fruit to CS, the TPM method was used to analyze the gene expression levels in the two stages (Fig. S1). The correlation coefficient between the three biological replicates was 0.87 in HJGA, 0.98 in TN1A, 0.96 in HJGB, 0.99 in TN1B, and the average correlation coefficient value was 0.95 (Fig. S2), indicating that the reproducibility of this study was good and the experimental results were reliable.

The software package DESeq2 was used to perform differential expression analysis of unigenes. There were 3,248 and 4,340 DEGs at two stages, respectively. After, analysis of the DEGs for the two stages, we found that the DEGs between HJG and TN1 were increased by 33.6% under CS condition (1,092), and 87.5% (955) were up-regulated (Fig. 2A).

Cluster analysis of gene expression can intuitively reflect the level of gene expression and expression patterns in multiple samples. We used the DEGs to perform cluster analysis on A1 vs A2 (Fig. 2B) and B1 vs B2 (Fig. 2C). The results showed that the difference between the three biological replicates of each group was small, which again confirmed the rationality of sample selection.

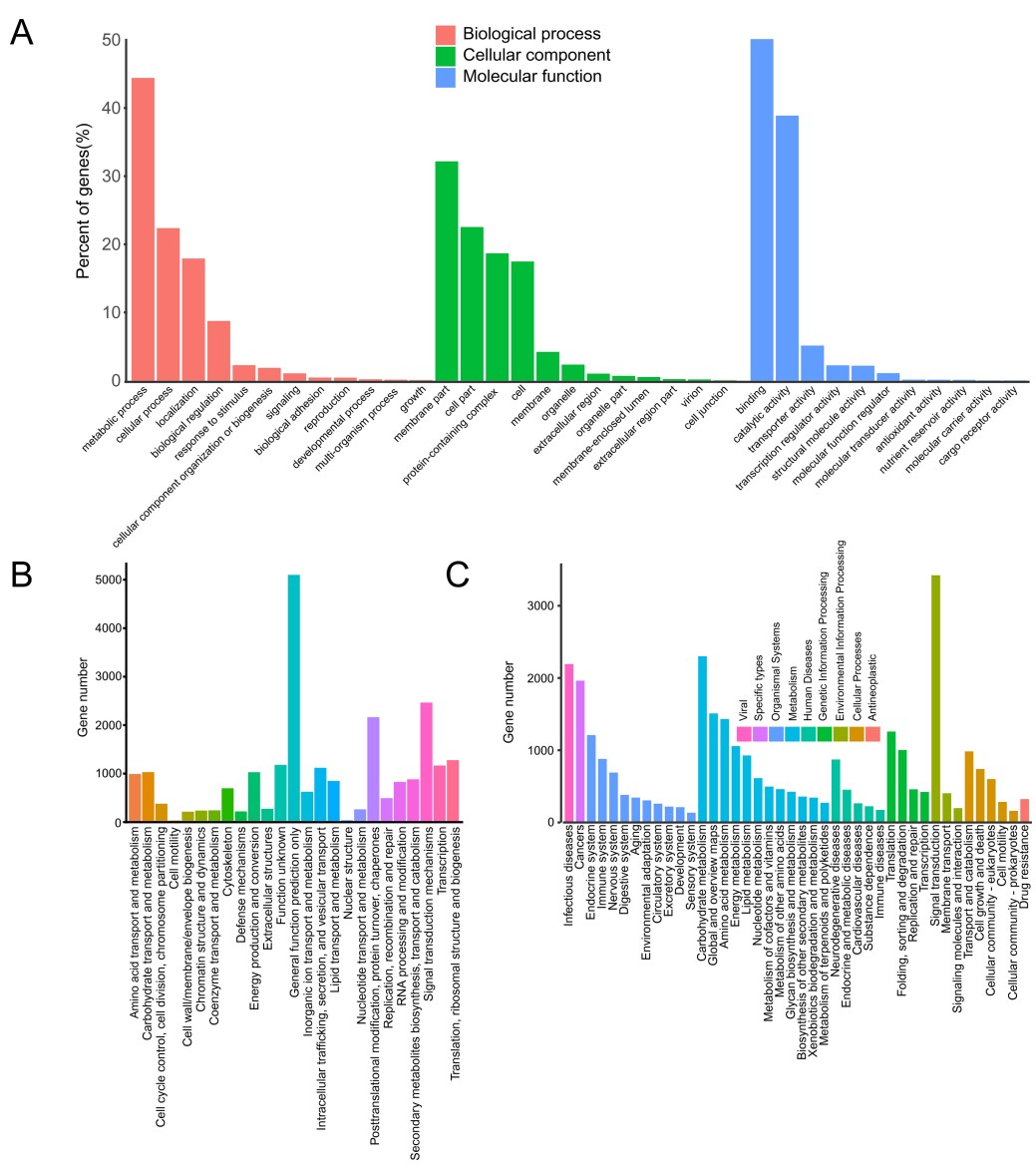

**Figure 1  Annotation of passion fruit transcriptome.** (A) GO function classification diagram of uni-genes. The *x*-axis indicates the secondary classification terms of GO; the *y*-axis indicates the number of unigenes in this secondary classification out of the total annotated unigenes. (B) KOG functional annota-tion distribution of unigenes. The *x*-axis indicates the nuber unigens; the *y*-axis indicates the name of 25 groups. (C) KEGG classification of unigenes. The *x*-axis indicates the number of unigenes in the pathway; the *y*-axis indicates KEGG pathways.

## GO and KEGG pathway enrichment analysis of DEGs

There were 1,182 up-regulated unigenes, and 2,066 down-regulated unigenes at stage A; and there were 2,137 up-regulated unigenes and 2,203 down-regulated unigenes at stage B.

GO enrichment analysis indicated that "metabolic process" (542), "oxidation–reduction process" (156), "protein phosphorylation" (92), "carbohydrate metabolic process"(73), "organic substance catabolic process" (40), and "catabolic process" (40), "extracellular

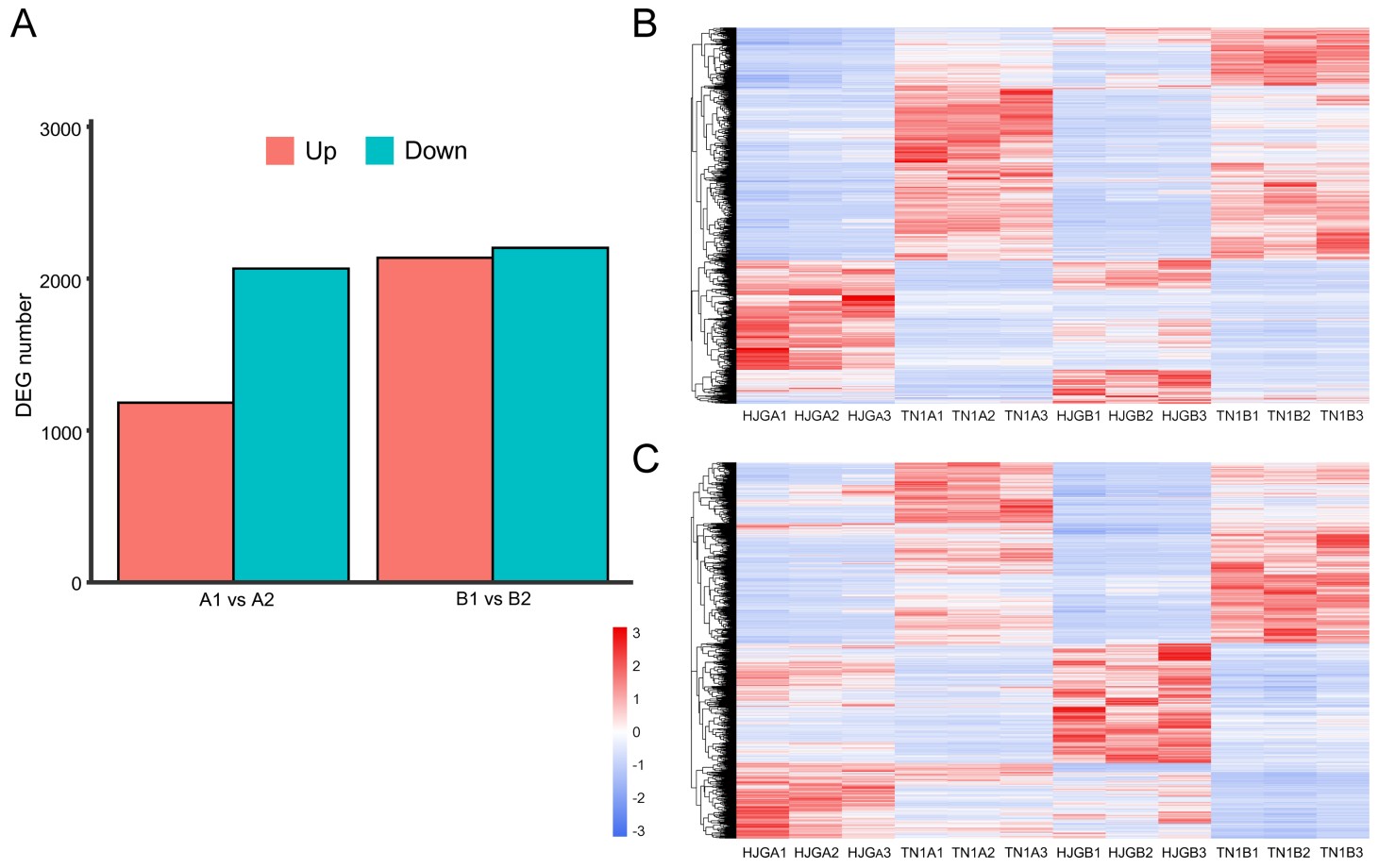

**Figure 2** **Analysis of DEGs at two stages.** (A) DEGs identified between HJG and TN1. (B) A1 vs A2; (C) B1 vs B2. Red indicates that the gene is highly expressed in the sample; blue indicates lower expression, and the number label under the color bar at the upper left is the specific trend of the change of expression. The dendrogram of gene clustering, and the name of the samples are indicated in the figure.

region" (10), "apoplast" (8), "cell wall" (8), "external encapsulating structure" (8), "catalytic activity" (634), "transferase activity" (228), "oxidoreductase activity" (167), "metal ion binding" (146), "cation binding" (146), and "transition metal ion binding" were enriched at stage A (110) (Table S2). But "oxidation–reduction process" (187), "phosphate-containing compound metabolic process" (171), "phosphorus metabolic process" (171), "macromolecule modification" (170), "cellular protein modification process" (169), and "protein modification process" (169), "membrane" (213), "intrinsic component of membrane" (99), and "integral component of membrane" (97),"catalytic activity" (837), "transferase activity" (326), "cation binding" (202), "metal ion binding" (201), "oxidoreductase activity" (198), "phosphotransferase activity", and "alcohol group as acceptor" (165) were enriched at stage B (Table S3).

The GO terms in A ($P > 0.05$) were compared to B ($P \leq 0.05$), and the unigenes were mainly about "protein phosphorylation" (GO:0006468 , 61) "phosphorylation" (GO:0016310, 61), "response to stimulus" (35), "lipid metabolic process" (19), "response
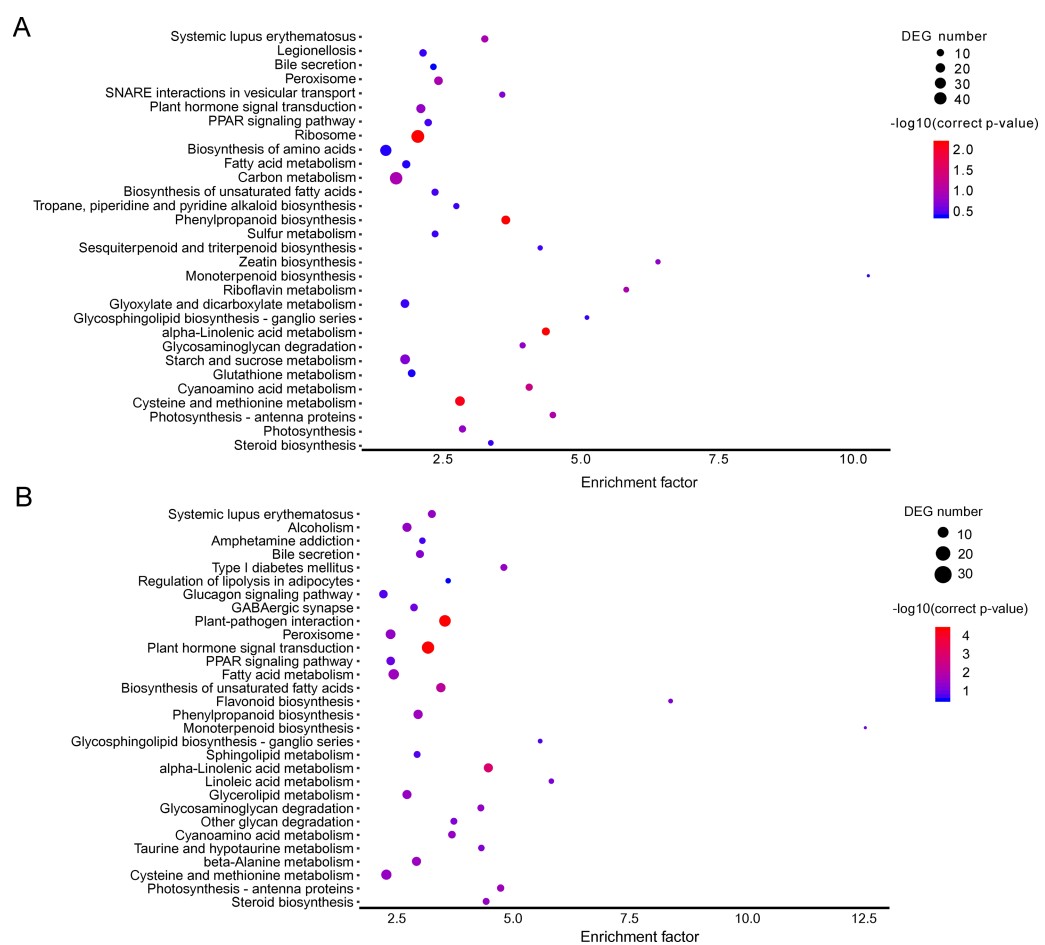

**Figure 3   KEGG pathway enrichment of DEGs.** (A) A1 vs A2. (B) B1 vs B2. The names of pathways, and enrichment factor are shown. The size of the dots indicate the number of genes in this pathway, and the color of the dots corresponds to different −log10(correct *p* value) ranges.

to chemical'' (13), ''membrane'' (73), ''intrinsic component of membrane'' (29), ''integral component of membrane'' (28) ''catalytic activity'', ''acting on a protein'' (77), ''transferase activity'', ''transferring phosphorus-containing groups'' (70), ''kinase activity'' (67), ''phosphotransferase activity'', ''alcohol group as acceptor'' (67), and ''protein kinase activity'' (62) (Table S4).

The KEGG pathway enrichment analysis can reveal the main metabolic pathways and signal transduction pathways in which the DEGs were involved, and the prevailing pathways were as follows: ''ribosome'' (42), ''carbon metabolism'' (39), ''biosynthesis of amino acids'' (30), ''starch and sucrose metabolism'' (21), ''and cysteine and methionine metabolism'' (20) at stage A; ''plant hormone signal transduction'' (31), ''plant-pathogen interaction'' (27), ''fatty acid metabolism'' (21), ''cysteine and methionine metabolism'' (20) (Fig. 3A). The KEGG pathway in A ($P > 0.05$) were compared to B ($P \leq 0.05$), and the unigenes were mainly about ''plant-pathogen interaction'' (17), ''plant hormone signal transduction'' (14), and ''fatty acid metabolism'' (8) (Fig. 3B).

## WGCNA analysis

After background correction and normalization of the unigenes expression data, we filtered out the abnormal and minor changed unigenes. Finally, we obtained 12,471 highly expressed unigenes (Table S5). In this study, when the soft threshold was 16 (Fig. S3), the gene topology matrix expression network was closest to the scale-free distribution. A gene cluster tree was constructed based on the correlation between genes, and each branch corresponded to a cluster of gene sets with highly correlated expression levels (Fig. S4A).

According to the standard of mixed dynamic shear, the gene modules were classified and the eigenvector of each module was calculated. The modules close to each other were merged, and 8 co-expression modules were obtained (Fig. S4B). Each module used different colors to represent the clustered genes. The turquoise module had the most clustered genes (4,171), the red module contained the fewest (81), and the grey module contained the unigenes that couldn't be included in any module.

The DEGs were used to draw the heat map of each module in the four sample groups. The brown and yellow modules showed less changes in differential unigenes between the early and late HJG, but showed larger changes in differential unigenes between early and late TN1 (Fig. 4), which is consistent with the chilling resistance feature of TN1. Therefore, we selected the unigenes of these two modules for in-depth GO and KEGG pathway analysis.

In the brown module, the significant GO terms were "cellular macromolecule metabolic process", "phosphate-containing compound metabolic process", "phosphorus metabolic process", "protein phosphorylation", "stimulus", "transferase complex", "riboflavin synthase complex", "photosystem I reaction center", "photosystem I" "binding", "metal ion binding", "cation binding", "phosphotransferase activity", "alcohol group as acceptor", "kinase activity" (Table S6). In the KEGG pathway analysis, the prevailing pathways were "plant hormone signal transduction", "MAPK signaling pathway", and "starch and sucrose metabolism" (Fig. 4A).

In the yellow module, the significant GO terms were "cellular process", "macromolecule modification", "phosphorus metabolic process", "cellular protein modification process", "protein modification process", "cell periphery", "photosystem", "photosynthetic membrane", "thylakoid", "extracellular region", "3-deoxy-7-phosphoheptulonate synthase activity", "alkylbase DNA N-glycosylase activity", "DNA-3-methyladenine glycosylase activity", "DNA N-glycosylase activity", and "method adenosyltransferase activity" (Table S7). The significantly enriched pathways included "biosynthesis of amino acids", "plant hormone signal transduction", "ABC transporters", "starch and sucrose metabolism", "folate biosynthesis" and "other pathways" that might be related to CS (Fig. 5B).

The constructed network was visualized with the Cytoscape in the brown and yellow modules, and got 19 hub unigenes which mainly related to "MAPK signaling pathway", "plant hormone signal transduction", "starch and sucrose metabolism", "fatty acid biosynthesis" and "photosynthesis in the brown module" (Fig. 6A). In the yellow module, 13 hub unigenes were mainly related to "plant hormone signal transduction", "MAPK signaling pathway", "starch and sucrose metabolism", and "fatty acid degradation" (Fig. 6B).

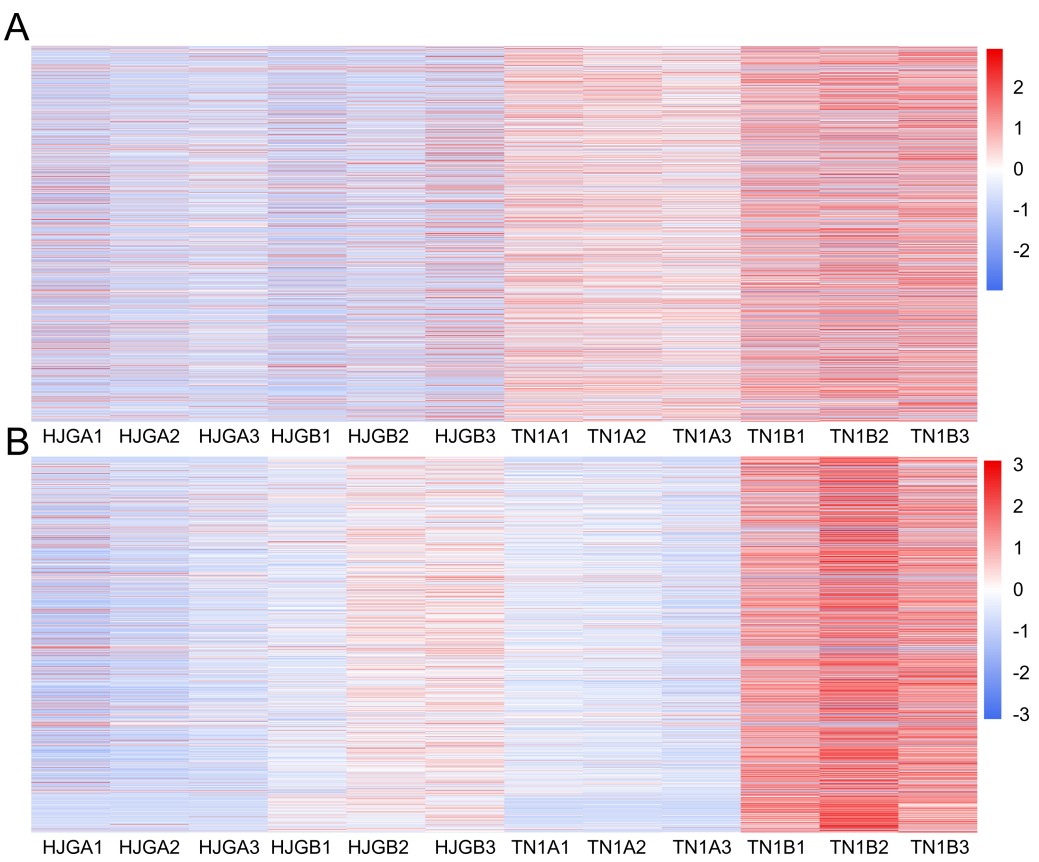

**Figure 4   Thr heat map of DEGs at two stages.** (A) A1 vs A2. (B) B1 vs B2. The below is the name of the samples. Red indicates that the gene is highly expressed in the sample; blue indicates lower expression, and the number label under the color bar is the specific trend of the change of expression.

## Validation of gene expression changes during chilling acclimation

We used the RT-qPCR method to validate the expression levels of 15 unigenes. The results showed that the RT-qPCR expression patterns of the 15 unigenes were consistent with RNA-seq analysis (Fig. 7, Table S8). RT-qPCR analysis showed that the 9 unigenes were ≥2 or ≤0.5 fold-change. Comparison with B1 and B2, TPM value of 12 unigenes were ≥2 or ≤0.5 fold-change. The results showed that seventy-five percent DEGs could be validated using RT-qPCR, and DEGs analysis were highly reliable.

## DISCUSSIONS

Low temperature is one of the main abiotic stresses that the plants are vulnerable to during their life cycle, and the response of plants to low temperature stress is a multi-factor synergistic process involving complex physiological and gene expression regulatory networks. With the development of molecular biology technology, researchers have cloned many low temperature related genes in *Arabidopsis thaliana* (*Wang et al., 2019*; *Ding et al., 2018*; *Ye et al., 2019*) and rice (*Ma et al., 2015*; *Zhang et al., 2017a*). Passion fruit is a tropical and subtropical fruit tree and it is vulnerable to low temperature in winter. However, there

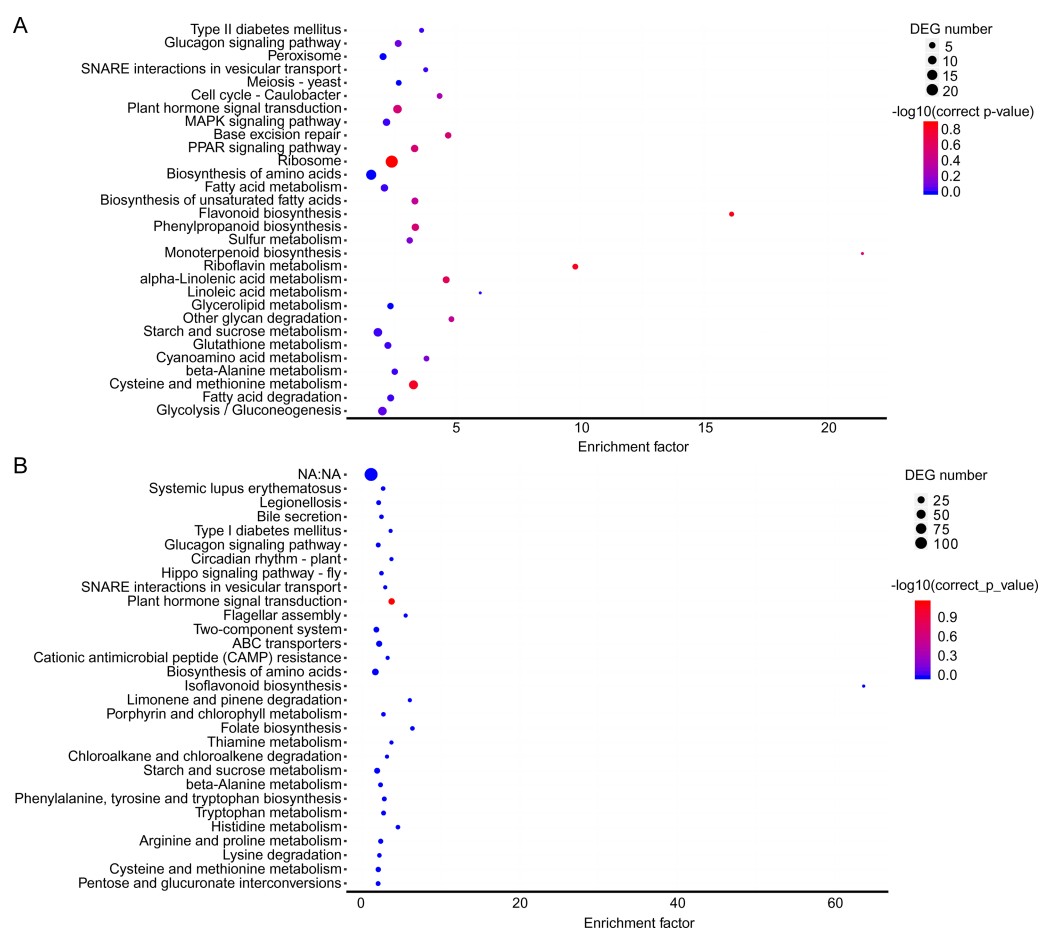

**Figure 5** **KEGG pathway enrichment in two co-expression modules.** (A) Brown module. (B) Yellow module. The *x*-axis indicates the enrichment factor; the *y*-axis indicates the name of pathways. The size of the dots indicate the number of genes in this pathway, and the color of the dots corresponds to different −log10 (correct *p* value) ranges.

are fewer studies on cold stress in passion fruit. In this study, the passion fruit cultivar of TN1 was identified, which has the characteristics of cold-tolerance.

Although the two cDNA libraries were constructed for transcriptome sequencing in passion fruit under CS condition (*Liu et al., 2017a*), we still know little about the cold tolerance in passion fruit. To reveal the expression pattern of CS-related genes in passion fruit, RNA-seq analysis were performed. Using database function annotation, we obtained 47,353 unigenes (https://figshare.com/s/89666a7f0fe7d4df353e). Based on RNA-seq data, the number of down-regulated DEGs did not change much at two stages, but the number of up-regulated DEGs were 955, indicating that the up-regulation of DEGs maybe related to CS.

Protein phosphorylation is also a type of post-translational regulation during the cold acclimation in plant. Under cold condition, CRPK1 is activated and phosphorylates 14-3-3λ, and the phosphorylated 14-3-3λ enters nucleus from the cytoplasm and degrades

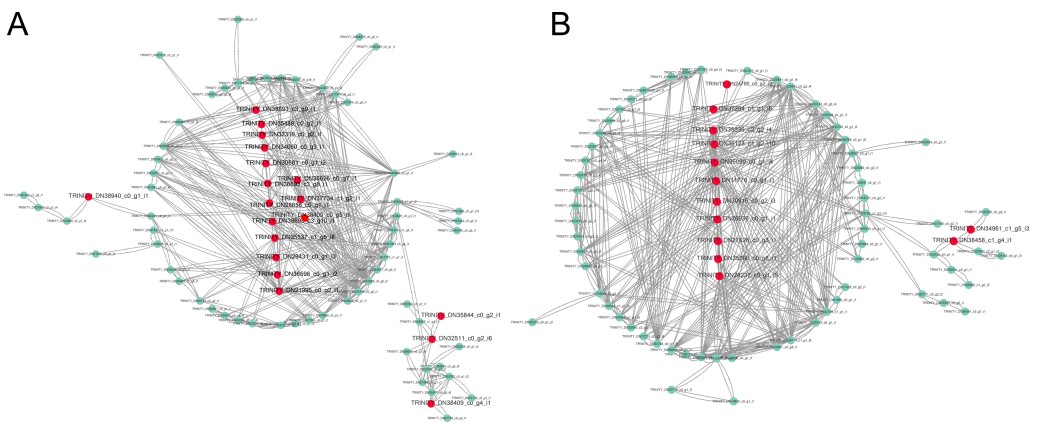

**Figure 6 Gene coexpression network related to cold stress.** (A) Gene co-expression network related to cold stress in brown module. (B) Gene co-expression network related to cold stress in yellow module. Red dots represent the hube gene belonging to the co-expression network.

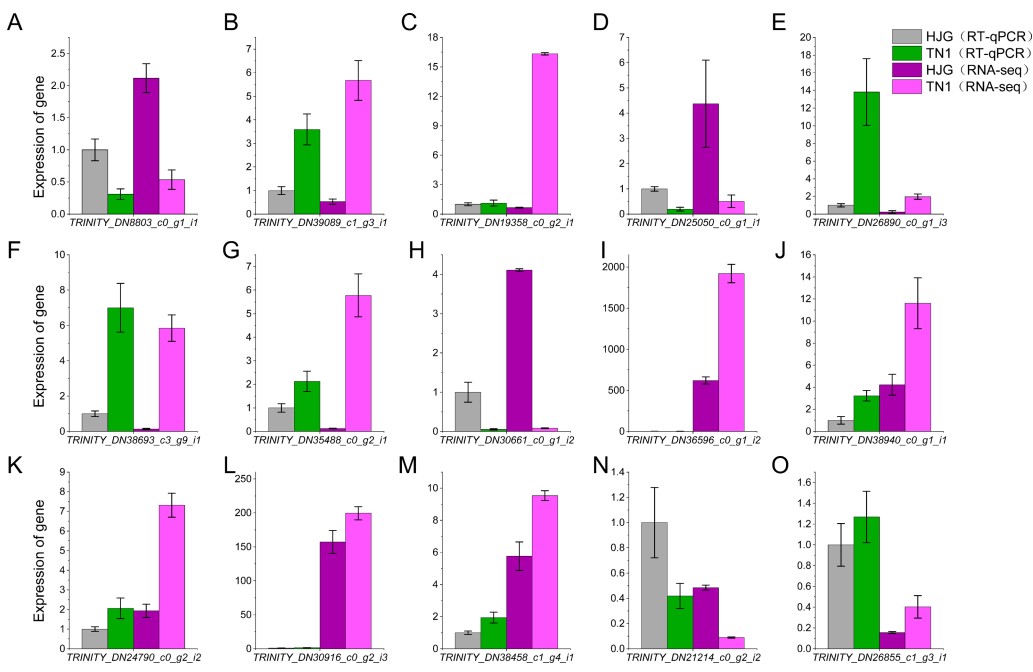

**Figure 7 Cold acclimation related genes (A–O) were validated by RT-qPCR.** The blocks indicate the samples of HJG and TN1 using in RT-qPCR and RNA-seq under cold stress condition. Bars indicate standard deviations of three biological repetitions.

CBFs via direct interaction in *Arabidopsis thaliana* (*Liu et al., 2017a*). Transcriptome sequencing revealed that 61 DEGs of phosphorylation were significantly up-regulated or down-regulated in the two stages, respectively (Table S4). Furthermore, the unigenes were mainly related to calcium-dependent protein kinase, serine/threonine-protein kinase, and CBL-interacting serine/threonine-protein kinase. In rice, calcium-dependent protein kinase

genes *OsCPK17* (*Almadanim et al., 2018*), *OsCDPK7* (*Saijo et al., 2000*) and *OsCPK24* (*Liu et al., 2018*) all respond to low temperature. In a previous study, serine/threonine protein kinase responded to cold stress (*Soto et al., 2002*). These results indicated that protein phosphorylation could play an important role in cold acclimation of passion fruit.

Mitogen-activated protein kinase (MAPK) plays an important role in signal transduction, and is also essential for regulating the cold response of plants. Under low temperature, the phosphorylation levels of MPK3, MPK4 and MPK6 were significantly increased (*Zhao et al., 2017*); moreover, MPK3 and MPK6 could interact with ICE1 to participate in low temperature response (*Li et al., 2017*). Zhang et al. found that the phosphorylated OsICE1 could promote *OsTPP1* transcription and induce the production of large amounts of trehalose, thereby improving the cold resistance of rice (*Zhang et al., 2017b*). Using WGCNA analysis, we found that MAPK signaling pathway significantly enriched in the brown module, which contained 7 DEGs (Fig. 5A). Moreover, the functional annotation of *TRINITY_DN36339_c2_g1_i5* was mitogen-activated protein kinase kinase kinase 3. In rice, OsMKK6 and OsMPK3 constitute a moderately low-temperature signalling pathway and regulate cold stress tolerance (*Xie, Kato & Imai, 2012*). MKK2 induces the expression of *COR* genes to enhance the freezing tolerance of *Arabidopsis thaliana* (*Teige et al., 2004*).

In plants, hormones and cold signaling pathways are coordinated to better adapt to cold stress. ABA is used as an important signal molecule and the most important stress signal in hormones, and it can mediate the signal transduction pathway to cold stress and increase the tolerance of cold stress (*Yuan, Yang & Poovaiah, 2018*). Auxin acts as a trigger in plant growth and development. In rice, ROC1 can regulate *CBF1*, and auxin can affect ROC1 levels (*Dou et al., 2016*). In addition, BR, GA, JA, ethylene, CK, and melatonin play important regulatory roles in the ICE–CBF–COR pathway (*Wang et al., 2017*). In CS condition, we found 31unigenes about plant hormone signal transduction (Fig. 3A). In WGCNA analysis, the pathway of plant hormone signal transduction was enriched in brown and yellow modules. These unigenes were annotated about aux, JA, ABA, and BR.

Plants use fatty acid dehydrogenase to regulate the increase of fatty acid unsaturation to improve the cold resistance (*Upchurch, 2008*; *He et al., 2015*). The change of malondialdehyde content caused by lipid peroxidation is negatively correlated with plant cold resistance (*Kim & Tai, 2011*). In this study, the unigenes related to fatty acid metabolism and lipid metabolic process were identified (Fig. 3A, Fig. 6B). Among them, 16 unigenes were annotated as delta (12)-fatty-acid desaturase (FAD2). In rice, *OsFAD2* is involved in fatty acid desaturation and maintenance of the membrane lipids balance in cells, and could improve the low temperature tolerance (*Shi et al., 2012*). Similarly, *FAD2* could improve the salt tolerance during seed germination and early seedling growth (*Zhang et al., 2012*), but *FAD8* was strongly inducible by low temperature in *Arabidopsis thaliana* (*Gibson et al., 1994*). The results indicated that *FAD2* could improve the CS of passion fruit.

In the process of cold acclimation in plants, the hydrolysis of starch is intensified and the content of soluble sugar increases. As a result, the freezing point of cell fluid is lowered and the excessive dehydration of cells is reduced (*Krasensky & Jonak, 2012*; *Yue et al., 2015*). The analysis of pathway enriched by KEEG and WGCNA revealed starch and sucrose

metabolism related to cold stress was enriched at stage B. Three DEGs were obtained at stage B compare to stage A, and these unigenes were annotated as beta-glucosidase and glucan endo-1,3-beta-glucosidase 3-like genes.

## CONCLUSIONS

In this study, we performed a comprehensive comparative transcriptome analysis between two passion fruit cultivars, to identify the gene expression level and analyze molecular mechanism of CS. This work showed that the unigenes of protein phosphorylation, MAPK signaling, plant hormones and fatty acid metabolism play important roles in the chilling tolerance between the two passion fruit cultivars. Furthermore, 32 hub unigenes were assigned to two modules, which could play a vital role in the chilling acclimation of passion fruit. In all, these findings provide a deepened understanding of the molecular mechanism of cold stress and could facilitate the genetic improvement of chilling tolerance in passion fruit.

## ACKNOWLEDGEMENTS

The authors thank to Dr. Yinghua Pan for help providing data analysis suggestions.

### Funding

This work was supported by the Guangxi Natural Science Foundation of China (2018GXNSFBA281024, 2019GXNSFAA245002), the National Natural Science Foundation of China (32060660), and the Guangxi's Ministry of Science and Technology (AB18294007), Guangxi Academy of Agricultural Sciences (2018YT19, TS2016010). The funders had no role in study design, data collection and analysis, decision to publish, or preparation of the manuscript.

### Grant Disclosures

The following grant information was disclosed by the authors:
Guangxi Natural Science Foundation of China: 2018GXNSFBA281024, 2019GXNS-FAA245002.
National Natural Science Foundation of China: 32060660.
Guangxi's Ministry of Science and Technology: AB18294007.
Guangxi Academy of Agricultural Sciences: 2018YT19, TS2016010.

### Competing Interests

The authors declare there are no competing interests.

### Author Contributions

- Yanyan Wu performed the experiments, analyzed the data, authored or reviewed drafts of the paper, and approved the final draft.

- Weihua Huang and Qinglan Tian performed the experiments, prepared figures and/or tables, and approved the final draft.
- Jieyun Liu performed the experiments, analyzed the data, prepared figures and/or tables, and approved the final draft.
- Xiuzhong Xia analyzed the data, prepared figures and/or tables, and approved the final draft.
- Xinghai Yang and Haifei Mou conceived and designed the experiments, authored or reviewed drafts of the paper, and approved the final draft.

## Data Availability

Raw data are available at NCBI: PRJNA634206.

The assembled transcripts and unigenes are available at Figshare:

Yang, Xinghai (2020): Transcript data for passion fruit chilling stress. figshare. Dataset. Available at https://doi.org/10.6084/m9.figshare.13489869.v2

Yang, Xinghai (2020): Unigenes of passion fruit chilling stress. figshare. Dataset. Available at https://doi.org/10.6084/m9.figshare.13489863.v4

RT-qPCR data are available in the Supplemental Files.

## Supplemental Information

Supplemental information for this article can be found online at http://dx.doi.org/10.7717/peerj.10977#supplemental-information.

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
