# Peer review of "Comparative transcriptomic analysis reveals the cold acclimation during chilling stress in sensitive and resistant passion fruit (Passiflora edulis) cultivars"

_PeerJ, doi:10.7717/peerj.10977_

## Round 0.1 · original submission · Major Revisions

Please take into consideration the reviewer's comments and provide back a point-by-point rebuttal letter addressing those concerns.

Reviewer 1 ·

Basic reporting

The current manuscript entitled “Comparative transcriptomic analysis identifies key genes and regulatory mechanisms of Passiflora edulis in response to chilling stress” is well organized and the findings of this study will provide necessary information for the development of passion fruit with enhanced chilling tolerance. Although, similar results have been reported in a huge number of MS regarding cold tolerance and it is well known that sugars, hormones, and other signaling pathways, etc. play a vital a necessary role in cold tolerance in several plant species. Notably, this study will add additional information to the present knowledge and help passion fruit researchers to continue work at the molecular level. However, this manuscript needs great improvement. Below I have mentioned some minor and major concerns which will help authors to improve the presentation and scientific value of their work.

Experimental design

Check the below comments

Validity of the findings

Check the below comments

Additional comments

Introduction
 Line 56-59, add 1 or more references for these statements.
 Line 59, define LT. Defines abbreviations the first time they appeared in the text. Check the entire document for this issue. Line, 69, 74, 75, 76, etc.
 Line 64, define ICE1-CBF-COR as they appear first time here.
 Line 60-63, expand this paragraph. How cold stress mainly chilling temp affects plant growth and production; and how plant response and adapt to chilling stress concerning morphological, physiological, biochemical mechanisms.
 Further, the authors should discuss the importance of a transcriptome approach with vital experimental proofs.
 Line 79, in this study….during cold stress….. it should be chilling stress. Please be specific while using this term. Cold include both freezing and chilling. Better to use chilling stress in the entire manuscript, figures, tables, captions, etc. Carefully check the whole manuscript and fix this issue.
 Line 80, mention the full name of passion fruit varieties as Huangjinguo (HJG; cold-sensitive) and Tainong1 (TN1; cold-tolerant). After that, authors can use HJG and TN1 in the entire manuscript.
 Line 82, define DEGs.
 Overall, the introduction needs to be rewritten. The authors should emphasize on the objectives of their work and discuss the background accordingly. For instance, low temperature includes both chilling and freezing. According to the title and temperature used in this study (7 oC) is chilling stress. So, the authors should be consistent using this term. Please emphasized on chilling term instead of LT. However, you need to define it as a part of cold stress, after this, stay focused only on chilling stress. LT term must be replaced with CT/CS (chilling temp/chilling stress) in the entire manuscript including abstract, introduction, material and methods, results, discussion, figures, tables, and captions.
Material and methods
 Line 96, replace LT with CT or CS. You can choose any of these two terms and be consistent with it throughout the text.
 Line 86, in plant materials, authors should mention the source of these two genotypes.
 Line 103, A single library requires 1 ug of RNA… Replace “ug” with the symbol of micro “μg”.
 I suggest authors combine RNA extraction and transcriptome sequencing section and concise the detailed information of sequencing.
 Line 126-127, define abbreviations wherever applicable and available.
 Cite a suitable and recent reference for HMMER.
 Line 130, analysis of DEGs. It should be Enrichment analysis of DEGs. Some sentences are poorly written. Please rewrite them in a well-organized style.
 DESeq2, define it as the package of Bioconductor/R.
 Keep the abbreviations inside the parenthesis, not the elaborated form, e.g., TPM, DFR, FC, etc. Check the entire manuscript for this issue.
 Line 146, ddH2O should be ddH2O.
 Line 153, correct the abbreviation within parenthesis (WGCNA).
 Line 157, mention the version of Cytoscape.
 Line 141, section quantitative real-time PCR (142-150) should be transferred to the validation of the cold acclimation-related genes. Combine these two sections and update the title as “validation of the cold acclimation-related genes using qRT-PCR”.
Results and discussion
 Merge Fig 1-3 in one figure and present them as Fig 1A/B/C. Same goes for Fig. 4 and 5. Merge Fig 4 and 5 in one figure and present them as Fig 2A/B/C.
 Line 185, cell composition (CC). It should be cellular component (CC). Check the entire manuscript.
 Line 213, DESeq2 is a package, not a software.
 Merge Fig 8 and 9 in one figure and present them as Fig A/B.
 Line 343-381, authors should relate this part with their findings. Currently, it is just a piece of general information for cold tolerance and accumulation, and the connection is not clear with the results of this study. Similarly goes for Line 382-408, the connection is not clear. The authors should discuss their findings with previous reports. Do not just briefly explain the cold-related mechanisms. Discuss them to support your results.
 Overall, the discussion needs serious improvement to support the authors' findings.

The English language needs great improvement. Please revise the language from a native speaker.

Reviewer 2 ·

Basic reporting

The structure of the article is according to the required format by PeerJ.
The introduction includes sufficient background although it is somewhat inconsistent with the rest of the manuscript (see below). It must show more clearly the context.
Figures are relevant, high quality and appropriately labelled and described. Tables are appropriately labelled.

Experimental design

The research question is well defined and relevant, it is stated how the research fills an identified knowledge gap. The methods are described with sufficient detail and information to replicate, however, a statistical analysis is missing (Figure 11, for example) or not clearly indicated.

Validity of the findings

The statistical analysis of the data must be appropriately described in the “Materials and Methods” section and clearly showed in figures and tables. The conclusions seem to be a small summary of the results rather than an analysis and conclusion of the data found. I recommend rewriting this section, which have to be linked to original research question and limited to supporting results.

Additional comments

The manuscript entitled “Comparative transcriptomic analysis identifies key genes and regulatory mechanisms of Passiflora edulis in response to chilling stress ” by Wu et al. addresses a very interesting topic. The authors performed RNA-seq of two passion fruit varieties HJG and TN1. The first one is considers cold sensitive and the second one is cold resistant. Both varieties were study in two different season when the temperatures were 25ºC (NT, Normal Temperature) and 7ºC (LT, Low Temperature). They made a comparative analysis of the differentially expressed genes (DEGs) between the passion fruit varieties under NT and LT conditions. The RNA-seq data were deposited to National Center for Biotechnology Information (NCBI) Sequence Read Archive (SRA) by authors. Using different databases as GO, KOG and KEGG, authors acquired annotation information of the unigenes and divided them into hypothetical functional categories.

This study provides an interesting but limited catalog of gene expression profiles. The data about the DEGs under different temperature conditions are also interesting and possibly useful for further research, however, I consider that the analysis that authors did is too much general. Analytic tools as GO, KOG and KEGG give only putative and very general functions of genes, it data can be uninformative and confusing (for example, are the protein phosphorylation and phosphorylation categories actually different as indicated in line 381? or why the catabolic process category is not included into the metabolic process category? line 212-214). It is difficult to obtain conclusive data based on this type of general analysis.

I consider that the authors should make a better analysis of the RNA-seq data focusing on genes with known functions in the regulation of cold tolerance such as genes of the ICE1-CBF-COR pathway. What about the CBP and COR gene expression? In the “Introduction” and “Discussion” sections, they mentioned some of these genes but did not analyze them, they only speculate about the possible roles of the signaling pathways and biological process deduced by GO or KEGG analysis in the cold tolerance phenomenon. Curiously, more than half of the “Introduction” section is about the ICE1-CBF-COR pathway but it is never mentioned in the “Results” section. I think that the lack of a more specific analysis makes the manuscript very speculative.


Additional comments:

I think that a comparative transcriptomic analysis is not sufficient to identify key genes and regulatory mechanisms, it analysis only can identify changes in gene expression, other experiments must be performed to demonstrate whether these changes have a functional impact. I consider that the title and some parts of the manuscript (lines 45-48 and 76-78) should be modified so that it could be in according with the results.
In the “Results” section, the subtitle “Gene expression analysis” is not according with the content of this section, actually the authors are not talking about gene expression analysis.

Minor comments:

A1, A2, B1 and B2 first appear in the Abstract, but their meaning is specified later, in the “Materials and Methods” section. It should be specified the first time it appears in the manuscript.
Major and secondary classification terms of GO, KOG and KEGG should be clearly differentiated from the main text, using italics or quotation marks for example.
Line 112. Change “sequnces” to “sequences”.
Line 135. Change “Tripricates” to “Triplicates”.
Line 370. Change “reduce the damage caused by cold damage” to “reduce the damage caused by cold”
Line 410. Change “collectd” to “collected”

---

## Round 0.2 · Minor Revisions

Please take into consideration the comments of the reviewers and send a revised manuscript. We acknowledge the efforts in this amended version.

Reviewer 1 ·

Basic reporting

First of all, I would like to say thanks to the authors for revising the MS according to the proposed comments and suggestions. Notably, the MS has been improved, however, only a few corrections, as mentioned below. Please refer to the track changes file for the line number.

Experimental design

No comments.

Validity of the findings

No comments.

Additional comments

 Line 65, add a space between to and cold.
 Line 91-92, keep the abbreviations within the parenthesis, not the elaborated form. Check the entire manuscript.
 Line 313, Fig. 42A. It should be Fig. 2A.

Reviewer 2 ·

Basic reporting

No comments.

Experimental design

No comments.

Validity of the findings

No comments.

Additional comments

The authors have adequately addressed most of the reviewers comments. However, the manuscript still contains some conclusions not completely supported by the results. A comparative transcriptomic analysis reveals changes in gene expression between two or more conditions, showing which genes change significantly. This type of analysis also allows us to suggest whether a gene or genes could have a function in the biological process of interest (chilling tolerance) but it is not enough to determine whether these genes really have a function or not in this particular phenomenon. Additional studies must be carried out to determine the function of the genes and the molecular mechanisms involved. These inconsistent conclusions are found in: line 43-44 “This work was the first systematic study of the molecular mechanism of chilling tolerance in passion fruit”; line 293-294 “To reveal the molecular mechanisms of chilling acclimation of passion fruits under CS, RNA-seq and analysis were performed”; and line 356-358 “32 hub unigenes were assigned to modules and played a regulatory role in the chilling acclimation of passion fruit”. The authors should change theses sentences for others more appropriate to the type of study carried out.

The manuscript contains many typographic and drafting errors: line 31-32 “we still know little is known about how the passion fruit…” instead of “little is known about how the passion fruit…” ; line 52 “cold sress” instead of “cold stress” ; line 177 “(BP),, cellular” instead of “(BP), cellular” ; line 235 “Fianlly, and obtained” instead of “Finally, we obtained” ; line 256 “annlysis” instead of “analysis” ; line 311 “tempreture” instead of “temperature” ; line 352 “omparative” instead of “comparative”. In many cases, space between words is lost.

---

## Round 0.3 · Minor Revisions

Please take into consideration the reviewer’s comments and provide back a point-by-point rebuttal letter addressing those concerns.

In particular, attend to the following comments, and provide an annotated transcriptome as supplementary materials or please deposit the assembled and annotated transcriptome GenBank in the TSA https://www.ncbi.nlm.nih.gov/genbank/tsa/ database:

"The information provided is informative, yet there is no link to actual data. The transcriptome is basically described without any links to sequence information. The authors do point to an SRA resource which would only be a means for readers to re-do the evaluation of the report, creating different results, and possibly providing different assessments. The authors need to provide a resulting Trinity sequence set (12471) for which can be aligned to the annotations and assessments. This manuscript is in need of revision to create the appropriate links. A PDF markup is also provided which highlights areas in need of attention."

---

## Round 0.4 · Minor Revisions

The authors have not deposited their raw transcriptome data on Genbank as indicated in the author instructions https://peerj.com/about/author-instructions/#data-and-materials

Besides creating a bio project and depositing the raw Illumina data, it is also customary to deposit the result of the assembled and annotated transcripts obtained by the Trinity pipeline. This will allow that a user can identify, for example, the function of any sequence listed in Supplementary data table 5.

I could not identify a Passiflora edulis transcriptomic bio project on GenBank, only a genome project in progress. Therefore, this is a great opportunity for you to have this data deposited and to contribute to the community with this key information for future studies on this plant. Your work will be highly cited for sure.

Here are some links that may help you to guide you in the data deposition.

https://submit.ncbi.nlm.nih.gov

https://www.ncbi.nlm.nih.gov/geo/info/seq.html

https://academic.oup.com/database/article/doi/10.1093/database/bax008/3737827

Be advised that this is a requirement for further acceptance of your contribution.

---

## Round 0.5 · Major Revisions

In general there is a bit of confusion in the way the data is supplied and organized. I see that supplemental data has been included; however, I am not seeing where it is presented in the manuscript. There were two FASTA datasets, yet there was no descriptions to distinguish the two. As part of your descriptions a general figure is presented which only sees a summary leaving the reader no way to understand its contents. For instance, though some of the TRINITY assemblies are provided, they are not presented in a way to distinguish the module associations or database assignments (GO:, KEGG); these would support the figures and numbers presented for the annotations, and help readers validate the work. This may not be a total loss and may be able to be recovered with a suggested 'major revision' decision; some reviews favored rejection. There are still some typographical errors and missing grammar which are minor at this point, but still need attention. I have provided a markup version. It is highly advisable that proof-editing assistance be done beforehand as it allows reviewers to focus on the science, rather than wade through the language context and delay reviews. I will return this as needing major revision to clear up the data structure with better guidance for what is provided; in manuscript and as supplemental. Also, bioproject raw data is usually deposited at and archive such as NCBI SRA, and depending on the assembly quality some data for assemblies is also accepted; you should consider taking these steps before re-submission.

---

## Round 0.6 · Minor Revisions

I see pointers to data in the rebuttal letter, but there are no pointers in the actual manuscript; you have listed PRJNA634206 and assembly data in figshare; none of these are mentioned in the manuscript. You would need to make appropriate edits to the manuscript to blend them into you presentation and discussion of the data.

---

## Round 0.7 · accepted · Accept

Sufficient changes requested have been added to the latest manuscript. The manuscript appears suitable to move forward in the review process, please consider the manuscript accepted at this stage; final approvals will be reviewed. Congratulations on the work provided; it is hopeful that it will be of value to other research in the area of cold acclimation.